# Continuously controllable photoconductance in freestanding BiFeO$_3$ by the macroscopic flexoelectric effect

Rui Guo[1,2,6], Lu You [3,6], Weinan Lin[1], Amr Abdelsamie[4], Xinyu Shu[1], Guowei Zhou[1], Shaohai Chen[1], Liang Liu[1], Xiaobing Yan[2✉], Junling Wang [4,5✉] & Jingsheng Chen [1✉]

Flexoelectricity induced by the strain gradient is attracting much attention due to its potential applications in electronic devices. Here, by combining a tunable flexoelectric effect and the ferroelectric photovoltaic effect, we demonstrate the continuous tunability of photoconductance in BiFeO$_3$ films. The BiFeO$_3$ film epitaxially grown on SrTiO$_3$ is transferred to a flexible substrate by dissolving a sacrificing layer. The tunable flexoelectricity is achieved by bending the flexible substrate which induces a nonuniform lattice distortion in BiFeO$_3$ and thus influences the inversion asymmetry of the film. Multilevel conductance is thus realized through the coupling between flexoelectric and ferroelectric photovoltaic effect in freestanding BiFeO$_3$. The strain gradient induced multilevel photoconductance shows very good reproducibility by bending the flexible BiFeO$_3$ device. This control strategy offers an alternative degree of freedom to tailor the physical properties of flexible devices and thus provides a compelling toolbox for flexible materials in a wide range of applications.

[1] Department of Materials Science and Engineering, National University of Singapore, Singapore 117575, Singapore. [2] College of Electron and Information Engineering, Hebei University, Baoding 071002, China. [3] Jiangsu Key Laboratory of Thin Films, School of Physical Science and Technology, Soochow University, Suzhou 215006, China. [4] Department of Materials Science and Engineering, Nanyang Technological University, Singapore 639798, Singapore. [5] Department of Physics, Southern University of Science and Technology, Shenzhen 518055, China. [6]These authors contributed equally: Rui Guo, Lu You. ✉email: yanxiaobing@ime.ac.cn; jwang@sustech.edu.cn; msecj@nus.edu.sg

Flexoelectricity has attracted considerable attention in recent years, which is an electromechanical property referring to a coupling between a strain gradient and an electric polarization[1]. The flexoelectric effect is significantly enhanced when reducing material dimension into nanometer range, since the size of the strain gradients is inversely proportional to its relaxation length[2]. Because the strain gradient breaks the inversion symmetry, flexoelectricity allows the generation of electric response from lattice deformations in otherwise centrosymmetric dielectric materials[3], which possesses great application potential in electronic devices. Particularly, through ferro-elastic/electric coupling flexoelectric effect has introduced a variety of intriguing phenomena in oxide ferroelectric thin films, e.g., modification of the polarization or altering domain structures of ferroelectric $BiFeO_3$ (BFO), $BaTiO_3$, or $PbTiO_3$ thin films[4–8], and manipulation of the defect configuration and associated electronic functions in ferroelectric BFO thin films[9,10]. Recently, it was reported that the strain gradient can mediate the local photoelectric properties of strained BFO thin film through the flexo-photovoltaic effect[11,12]. In all those cases above, the strain gradient is generated by one of the following methods: (1) thin film deposition by the lattice mismatch and relaxation[1]; (2) by changing the deposition temperature[7,9]; (3) the crystallographic disorders in the as-grown films (such as the morphotropic phase boundaries)[8,11,13,14]; and (4) by applying an external force via a scanning probe microscope tip to the film[5,6,10,15]. However, the strain gradient generated by these methods either a localized effect or cannot be tuned continuously. To take fully advantage of the flexoelectric effect and broaden its applications, a universal and facile method of introducing controllable macroscale strain gradient in functional thin films becomes urgent and essential.

With the development of the fabrication techniques of flexible crystal materials, high-quality flexible devices have been achieved recently[16–20]. The most prominent feature of the flexible devices is their bendable character, which enables continuous tuning of the strain gradient by gradually changing the bending radius of the flexible thin films[21–24]. In this work, we demonstrate the tunable photovoltaic effect in a freestanding single-crystal BFO film transferred onto polydimethylsiloxane (PDMS) substrate. The photovoltage/photocurrent is switchable upon the ferroelectric polarization switching. Furthermore, multilevel photoconductance in BFO is obtained when altering the bending radius of the flexible device, which is attributed to the change of strain gradient with bending radius and thus the built-in electric field across the device. In principle, the photoconductance can be continuously tuned upon changing the device bending radius. The multilevel photoconductance shows very good reproducibility, which has the potential to be used for multilevel nonvolatile memories with electric/mechanical writing and optical reading, or strain sensors, etc. Our findings demonstrate a viable route to expand the device functionality via using the strain gradient as a degree of freedom in flexible electronics.

## Results

**Device fabrication and characterization of freestanding BFO.** Environmentally friendly multiferroic BFO thin film is chosen in this study due to its fascinating ferroelectric and photovoltaic properties[25–28]. It has a direct band gap within the visible light range (near 2.74 eV)[29] and a very large remnant ferroelectric polarization[30], which offers a unique opportunity for photovoltaic investigation and memory application. In this work, tunable strain gradient in BFO is achieved via mechanically bending the flexible device of Pt/BFO/$La_{0.67}Sr_{0.33}MnO_3$ (LSMO). Water-soluble $Sr_3Al_2O_6$ (SAO) is chosen as the sacrificing layer to fabricate the flexible ferroelectric device. SAO has a cubic lattice structure with the lattice constant $a = 15.844$ Å, which closely matches four-unit cells of $SrTiO_3$(STO; $a = 3.905$ Å)[16]. Freestanding crystalline oxide perovskite down to the monolayer limit, super-elastic ferroelectric single-crystal membrane, together

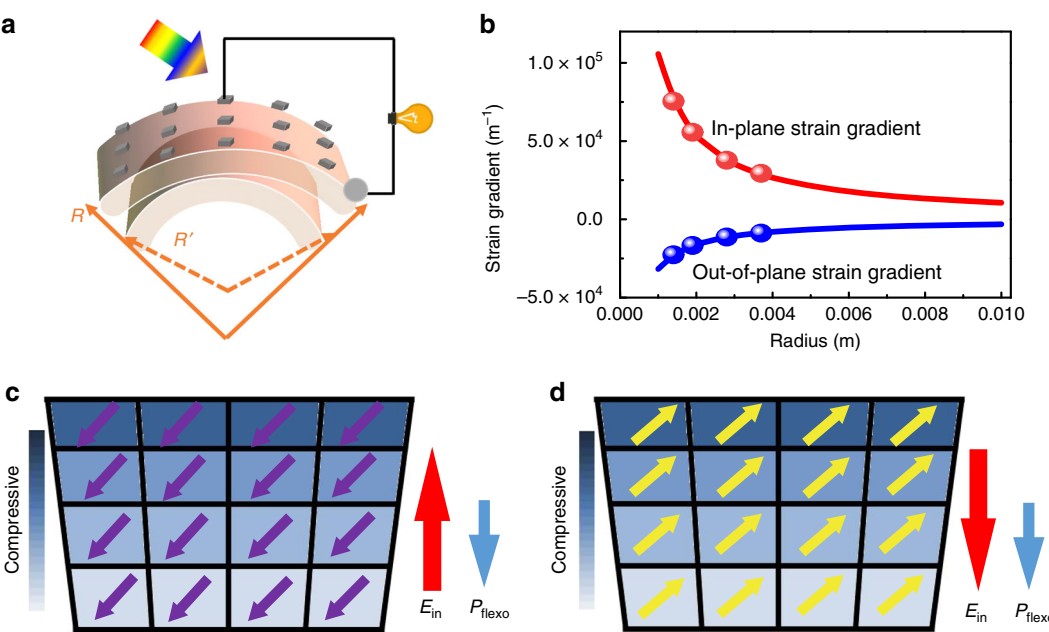

**Fig. 1 Bending induced strain gradient in the flexible device. a** Schematic of the realization of the strain gradient by bending the freestanding BFO thin film. **b** The solid lines are the calculated strain gradient as a function of the curvature radius. The eight solid points are the in-plane and out-of-plane strain gradients calculated from the four bending levels conducted in this work. Schematic of the stain status in the freestanding BFO thin film with **c** downward polarization and **d** upward polarization. The purple and yellow arrows represent the downward and upward BFO polarizations that are along the body diagonal direction, respectively. The red arrow represents the BFO polarization-dependent internal electric field ($E_{in}$) of Pt/BFO/LSMO device, and the blue arrow shows the flexoelectric polarization ($P_{flexo}$) generated by the flexoelectric effect during bending.

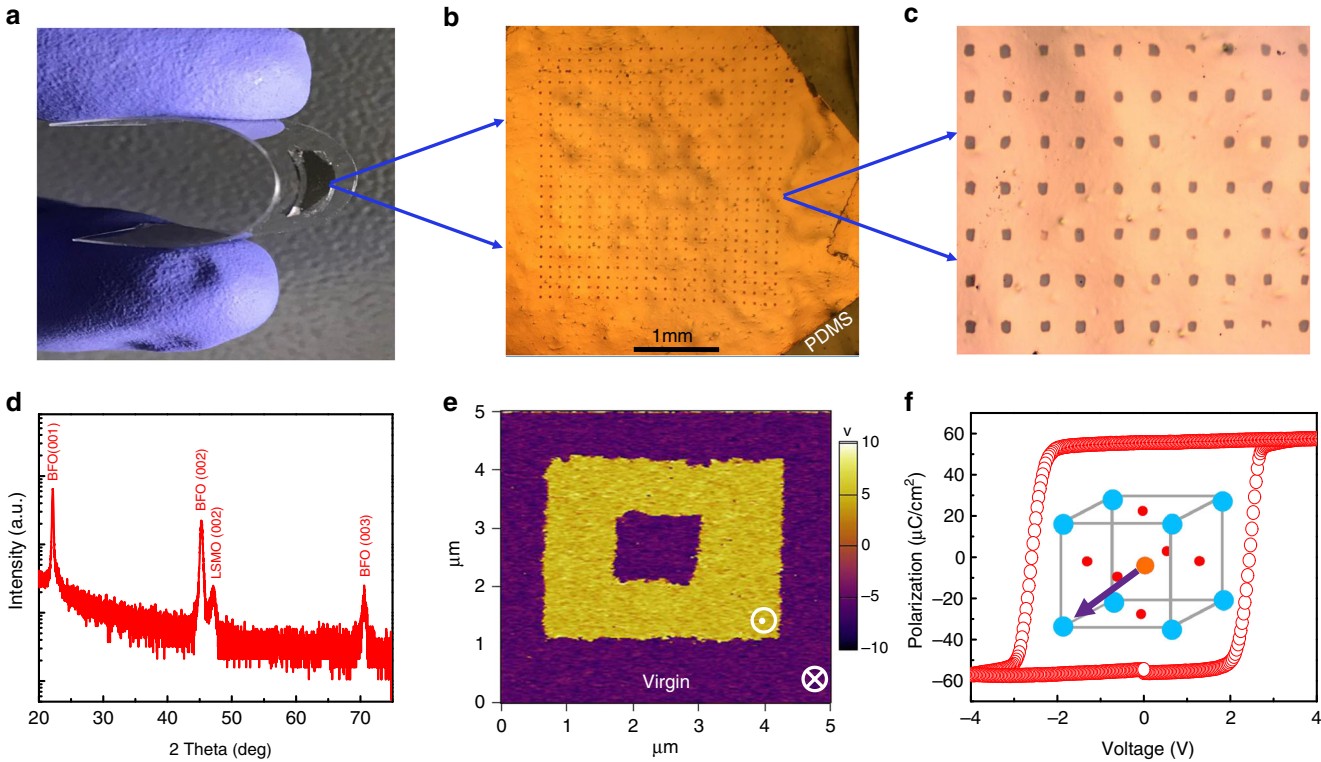

**Fig. 2 Fabricated flexible devices and basic characterizations of freestanding BFO.** Optical images of **a**, the bent Pt/BFO/LSMO sample and **b**, the flat one on PDMS supporter with patterned top Pt electrodes. **c** The local enlarged image of **b**. **d** XRD results of the freestanding BFO/LSMO thin films on Si supporter. **e** Out-of-plane PFM phase images of the freestanding BFO film, with the virgin state and the state after switching using a bias of −5 and +5 V, respectively. **f** P–V loop of the freestanding Pt/BFO/LSMO device. The inset shows the virgin single-domain polarization direction.

with other freestanding ferroelectric oxide memory devices have been demonstrated very recently using SAO as the sacrificing layer[31–34]. In our work, SAO was epitaxially grown on (001) STO (4° miscut toward (110)) single-crystal substrate followed by the deposition of 15 nm LSMO as the bottom electrode, and finally 100 nm BFO as the functional layer. Miscut STO substrates were chosen to obtain single-domain BFO films to eliminate complications from multiple domains in BFO. All the films were deposited by pulsed laser deposition (PLD) technique. After the deposition of thin films, the SAO layer was completely removed by simply immersing the samples into deionized (DI) water at room temperature, and then the isolated BFO/LSMO layer was transferred to PDMS substrates that are coated to polyethylene terephthalate (PET) supporter. Finally, an array of 50 μm × 50 μm Pt as the top electrodes was patterned for the electrical measurements. The flexible device fabrication process is illustrated in Supplementary Fig. 1. The schematic of the transferred freestanding Pt/BFO/LSMO device on PDMS is shown in Fig. 1a. Different bending states of the freestanding BFO can be indicated by the radius $R$ of the flexible device. Essentially, the magnitude of the strain gradient is determined by the bending radius of the device, with the data shown in Fig. 1b. Controllable strain gradient in freestanding BFO can thus be obtained by continuously bending the flexible device. Under an upward bending, a non-uniform lattice distortion is induced, with the strain status in freestanding BFO with different polarizations shown in Fig. 1c, d. The additional built-in electric field generated by the strain gradient through the flexoelectric effect will couple with the polarization-dependent internal field ($E_{in}$) to determine the photocurrent/photovoltage of BFO. Strain gradient therefore can be employed as a degree of freedom to tune the photovoltaic response to achieve multilevel conductance, using bendable freestanding Pt/BFO/LSMO.

Figure 2a, b shows the typical optical images of the bent Pt/BFO/LSMO sample and the flat one on PDMS with patterned Pt top electrodes, whose size can be as large as the STO substrate size (5 mm × 5 mm). Figure 2c displays the enlarged image of Fig. 2b. To carry out the structural and ferroelectric property characterizations, freestanding BFO/LSMO film was transferred to Si substrate after removing SAO, of which the scanning electron microscopy (SEM) images are shown in Supplementary Fig. 2. The smooth morphology of transferred freestanding BFO film implies that the transferring process retains the qualities of the as-grown films. The high-quality epitaxial films are further confirmed by the x-ray diffraction (XRD) results of the as-grown films with the sacrificial SAO layer (Supplementary Fig. 3a), and the transferred freestanding BFO/LSMO films on Si as shown in Fig. 2d. The surface topography and ferroelectric domain structure of the freestanding single-crystalline BFO were measured using piezoresponse force microscopy (PFM) technique. The step-bunching topography shown in Supplementary Fig. 3b results from the large substrate vicinality enforced film growth. Interestingly, a single downward domain structure of the freestanding BFO film is confirmed by the out-of-plane and in-plane PFM phase images, as shown in Fig. 2e and Supplementary Fig. 3c, respectively. The freestanding BFO has the same polarization direction as the as-grown one with SAO layer (Supplementary Fig. 4). This single-domain structure is due to the large miscut angle of (001) STO toward (110) that lifts the degeneracy of the multiple domains in BFO, resulting in the preferred ferroelectric polarization direction as indicated in the inset of Fig. 2f. Note that the preferred polarization direction of the freestanding film is different from that grown directly on the STO substrates (Supplementary Fig. 5). This is probably because that the grown SAO layer leads to a different termination of the following LSMO layer, which therefore induces different

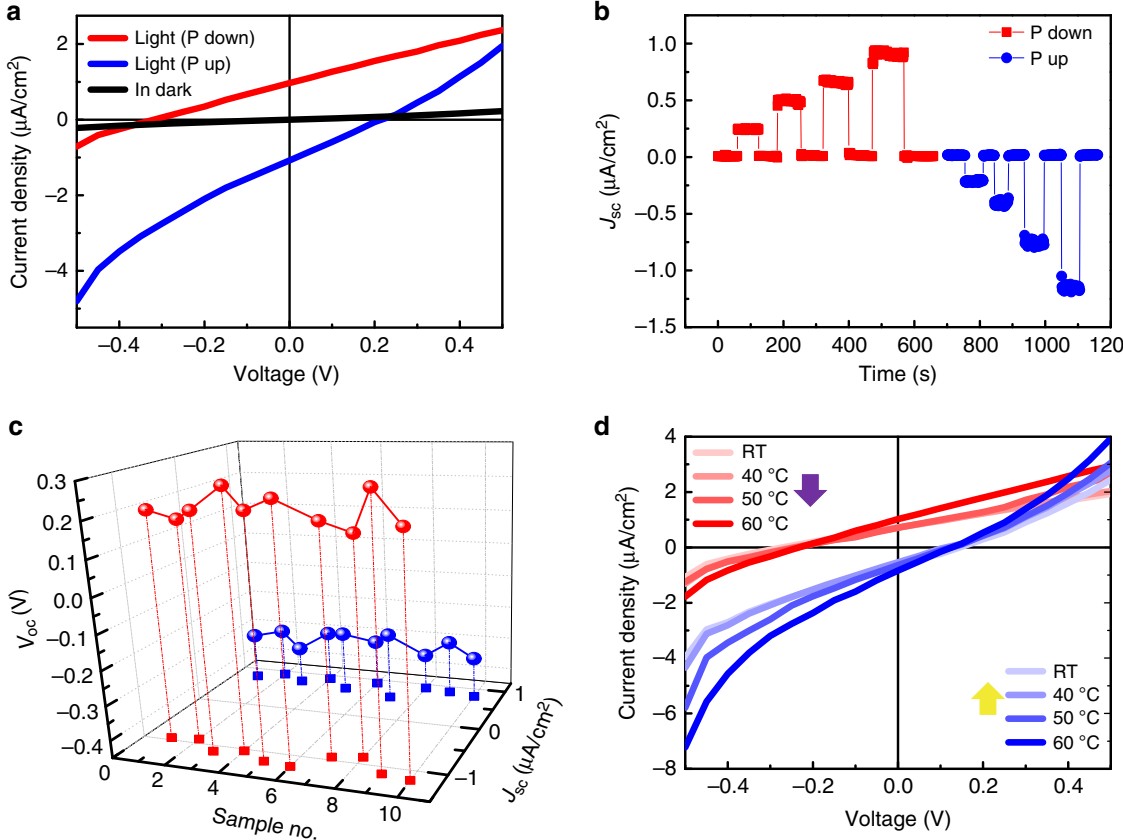

**Fig. 3 Photovoltaic properties of freestanding Pt/BFO/LSMO devices. a** *J–V* curves of the transferred freestanding Pt/BFO/LSMO photovoltaic cells under light illumination and in dark. Light source: halogen lamp with energy density of 20 mW/cm². **b** Photo response as function of time under different light intensities or in dark for both polarizations. $J_{sc}$ drops to 0 when turning off the light and increases with the increase of light intensity. **c** Uniformity of $V_{oc}$ and $J_{sc}$ of randomly chosen devices from different five freestanding photovoltaic samples. **d** Thermal stability of the freestanding photovoltaic devices with *J–V* curves measured for both polarizations at different temperatures from room temperature to 60 °C. The purple and yellow arrows represent the downward and upward polarization of BFO.

polarization directions of the subsequent ferroelectric BFO layer[35,36]. Furthermore, the ferroelectric polarization–voltage (*P–V*) hysteresis loop of the freestanding Pt/BFO/LSMO capacitor was characterized as shown in Fig. 2f. The *P–V* loop reveals a remnant polarization of around 60 μC/cm² along the $[001]_{pc}$ direction, which is consistent with the previous reports and indicates an excellent ferroelectric property of the transferred freestanding BFO thin films. The coercive voltage of the transferred BFO film is around ±2.5 V, as revealed in the *P–V* loop.

**Basic photovoltaic behavior**. Ferroelectric photovoltaic effect of the transferred Pt/BFO/LSMO was measured to verify the functionality of the flexible device. For the electrical measurements, the applied voltage is defined as positive (negative) if a positive (negative) bias is applied to the top Pt electrode. Current density–voltage (*J–V*) curves were measured under light illumination or in dark after switching the BFO polarization up or down by applying a bias of −5 or +5 V, as shown in Fig. 3a, from which the ferroelectric photovoltaic effect is demonstrated clearly. More importantly, ferroelectric polarization reversal of BFO thin film changes both the signs of open circuit voltage ($V_{oc}$) and short-circuited current density ($J_{sc}$), indicating that the switchable nature of photovoltaic effect is mainly attributed to the spontaneous polarization of BFO. The asymmetric $V_{oc}$ and $J_{sc}$ is likely due to the different work functions of the top and bottom electrodes, which also generates an internal field that does not depend on the polarization direction. Built-in

electric field $E_{in}$ is used to denote the polarization-dependent internal field that generates the ferroelectric photovoltaic effect. The polarization-dependent $V_{oc}/J_{sc}$ therefore can be used to read the polarization direction of BFO nondestructively by illuminating the devices. The optical reading method can also take advantage of the high speed of light, which would also be energy efficient if the light is simply visible light. Note that, the basic photovoltaic properties of the freestanding Pt/BFO/LSMO devices on PDMS display negligible difference from those of the rigid as-grown samples (Supplementary Fig. 6), proving the high quality of the transferred devices. Figure 3b shows the instantaneous response of $J_{sc}$ with time when switching the light on/off with different light intensities, which further confirms the steady photovoltaic effect of the freestanding ferroelectric devices. *J–V* curves under different light intensities were also measured as shown in Supplementary Fig. 7a–c, showing that the $V_{oc}/J_{sc}$ increases with the increasing light intensity. With the increase of light intensity, more photocarriers can be generated, which consequently leads to larger $V_{oc}$ and $J_{sc}$. Supplementary Figure 7d also demonstrates the *J–V* switching loops of freestanding Pt/BFO/LSMO under light illumination and in dark, which displays obvious hysteresis behavior and the photoresistance effect. Halogen lamp was used as the visible light source for the photovoltaic effect measurements, with the highest energy density of 20 mW/cm², which is 20% of the energy density of the sun. Therefore, a larger current density is expected with higher light intensity. Besides, both $V_{oc}$ and $J_{sc}$ could be improved by band structure engineering[37,38].

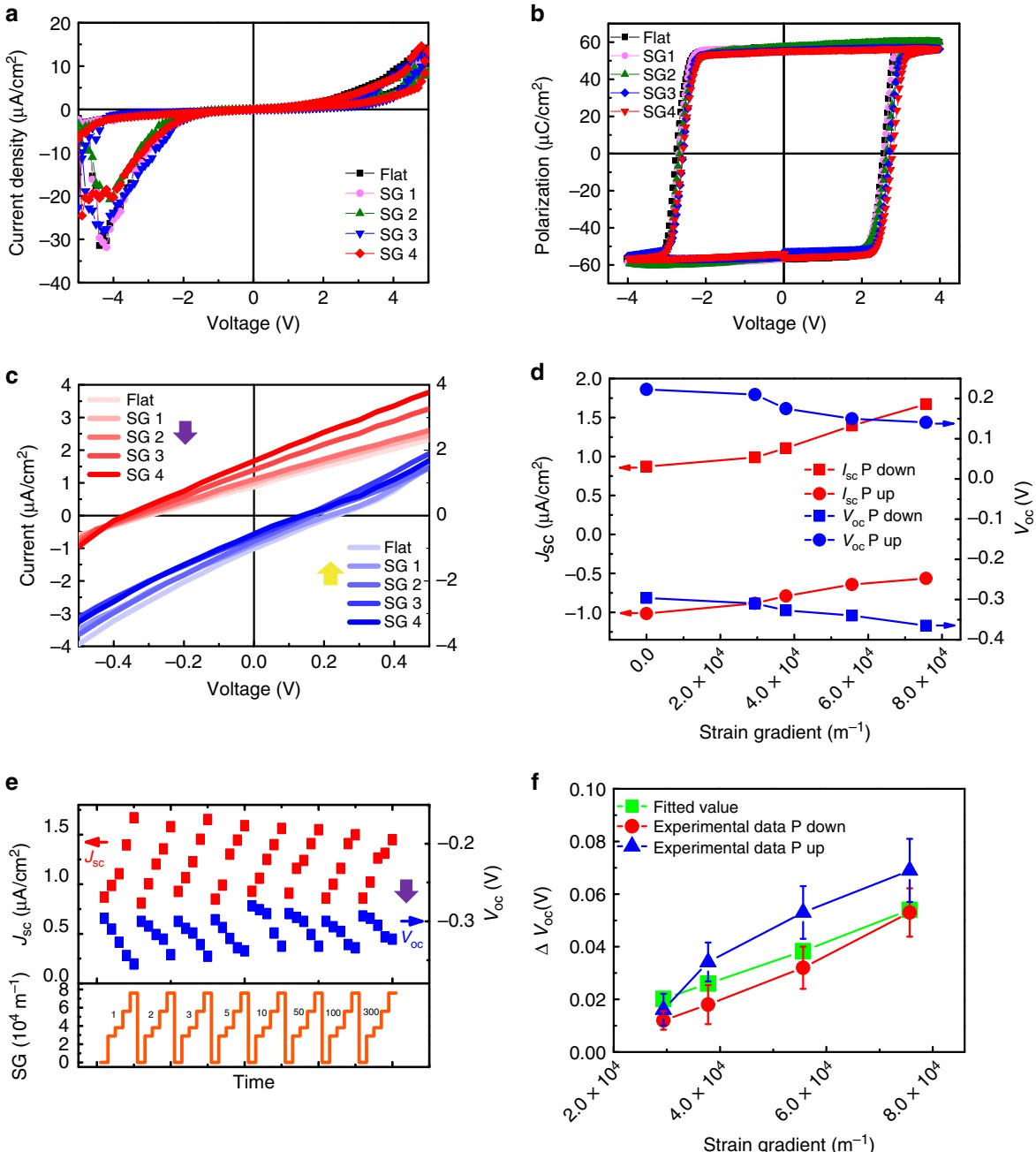

**Fig. 4 Bending-tunable photovoltaic properties of freestanding Pt/BFO/LSMO. a** $J–V$ switching loops in dark of the freestanding photovoltaic cells with different bending status. **b** $P–V$ loops of the flexible device as a function of the strain gradient. **c** $J–V$ curves under light illumination for both polarizations (with light intensity of 20 mW/cm²) of the flexible devices with different bending radius. Red curves: polarization downward; blue curves: polarization upward. **d** The corresponding change of $V_{oc}$ and $J_{sc}$ as function of the introduced in-plane strain gradient for both polarizations. **e** The change of the $V_{oc}$ and $J_{sc}$ with the function of the introduced in-plane strain gradient (SG) after different bending cycles. The polarization of BFO is downward. **f** The experimental data of $\Delta V_{oc}$ with both upward and downward polarization directions, and the fitted $\Delta V_{oc}$ as the function of the strain gradient by using the value of 0.5 for scaling factor λ. SG1 to SG4 corresponds to the increasing strain gradients of four different bending radii in Fig. 1b. The purple and yellow arrows in **c** and **e** represent the downward and upward polarization of BFO. The error bars in **f** are calculated from multiple measurement results in **e** and Supplementary Fig. 10h.

The uniformity of the freestanding devices was investigated further. Figure 3c shows the results of randomly chosen devices in five different samples. The $V_{oc}$ of the flexible devices with upward or downward polarization is around −0.3 and 0.2 V, respectively, while the $J_{sc}$ of the devices is on the order of μA/cm². The results indicate the good uniformity of the freestanding Pt/BFO/LSMO devices, which guarantees their functionality on arbitrary flexible substrates. Besides the uniformity, good thermal stability is also

important for flexible electronic devices. To test the thermal stability, $J–V$ curves under illumination were measured at different temperatures from room temperature to 60 °C. As shown in Fig. 3d, Pt/BFO/LSMO on PDMS retains good photovoltaic property even at 60 °C. Note that with the increase of the temperature, $J_{sc}$ increases with the increase of the temperature, while the $V_{oc}$ with both signs displays a slight decreasing trend. This is because that the increasing temperature

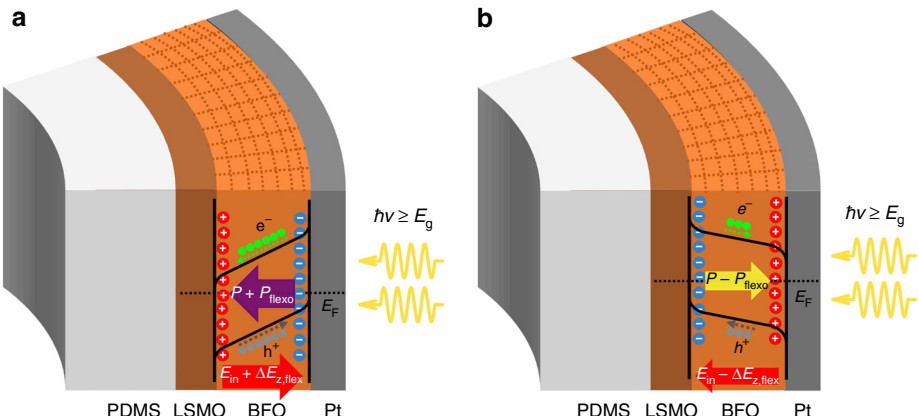

**Fig. 5 Tunable band diagram across the device by the flexoelectric effect.** The band bending diagram of BFO film at bent status with **a** downward polarization toward LSMO and **b** upward polarization toward Pt. Flexoelectric polarization $P_{flexo}$ points downward. When $\hbar v \geq E_g$ of BFO, electron–hole pairs will be separated by the internal built-in field ($E_{in}$). The additional electric field ($\Delta E_{z,flexo}$) generated by the flexoelectric field will enhance or weaken the photovoltaic effect depending on the BFO polarization direction.

leads to lower resistance of the insulating BFO layer, the leakage of which therefore slightly decreased the $V_{oc}$. Despite the weak influence of the temperature on $J_{sc}$ and $V_{oc}$, the stable photovoltaic effect at 60 °C proves the good thermal stability of the flexible devices. Furthermore, we investigated retention and endurance properties of the freestanding devices, shown in Supplementary Fig. 8, which are two critical requirements for memory applications. One can see that negligible deterioration in the signal was observed when the $V_{oc}$ and $J_{sc}$ were continuously monitored for 1 h after switching the polarization upward or downward. As for the endurance properties, the flexible devices have been subjected to bipolar switching for $10^5$ cycles under the switching bias of ±3 V with the pulse width of 1 ms. The slight deviation of the $J$–$V$ curves might be due to the fatigue that is possibly caused by plastic deformation or ferroelectric domain rotation, etc. Both the good photovoltaic properties, and the strong retention and endurance properties indicate that the transfer procedure has little impact on the properties of the freestanding ferroelectric devices. The functional freestanding Pt/ BFO/LSMO devices therefore can be applied as flexible memories that could solve the destructive reading problem of the conventional FeRAM.

**Bending-tunable photovoltaic behavior.** Then, we introduced strain gradient by bending the flexible device upward to investigate its tunable effect on the photovoltaic property of the freestanding BFO. The functionality of the flexible device under bent was investigated first, with the $J$–$V$ switching loops shown in Fig. 4a, which were measured on the original flat device and the bent device with different bending radii. It can be clearly seen that the freestanding Pt/BFO/LSMO device exhibits good resistive switching behavior, even at large bending status with bending radius of 1.4 mm. $J$–$V$ switching loops together with the $J$–$V$ curves within small voltage range were also measured after bending the devices multiple times, as shown in Supplementary Fig. 9, which reveals no deterioration in the photovoltaic property. $P$–$V$ loops as a function of the bending radius were then measured as shown in Fig. 4b. $P$–$V$ loops show a horizontal right shift with the increase of the strain gradient, which indicates the generation of an effective built-in electric field with bending. Through bending, sizable strain gradient is induced due to the nonuniform lattice distortion, leading to the additional effective built-in electric field via the flexoelectric effect that can tune the photovoltage/photocurrent of BFO. After bending the devices to different levels, $J$–$V$ curves within a small voltage range under

light illumination were measured accordingly. Figure 4c shows the typical $J$–$V$ curves of the flexible devices under light illumination with the change of the strain gradient induced by different bending status, and the extracted $V_{oc}$/$J_{sc}$ from the $J$–$V$ curves as a function of the strain gradient are summarized in Fig. 4d. One can see that when BFO polarization is switched downward, both $V_{oc}$ and $J_{sc}$ of the device increase with the increasing of the strain gradient. The $V_{oc}$ and $J_{sc}$ increase to −0.365 V and 1.67 μA/cm² with the in-plain strain gradient of 7.55E4, which have been enhanced by 24% and 90%, respectively, compared with those of the devices in the flat status (0.295 V and 0.88 μA/cm²). Such enhancement can be further improved with a larger strain gradient (smaller bending radius). On the contrary, both $V_{oc}$ and $J_{sc}$ decrease with the increasing of the strain gradient, when BFO polarization is poled upward. Mechanically bending the freestanding Pt/BFO/LSMO device to different levels consequently results in the multilevel photovoltage/photocurrent. More importantly, the continuously tuning of the photoconductance by controlling the strain gradient is repeatable. Multilevel photovoltage/photocurrent can still be obtained after bending the flexible device up to 300 times, as shown in Fig. 4e (P down) and Supplementary Fig. 10h (P up), which confirms the reproducibility and stability of the flexoelectrically tunable photovoltaic effect in freestanding BFO film. The averaged change of $V_{oc}$ ($\Delta V_{oc}$ or flexo-photovoltage) with both polarizations as a function of the strain gradient is summarized in Fig. 4f. The error bars are calculated from multiple measurement results in Fig. 4e and Supplementary Fig. 10h. The continuously tunable photovoltaic response in freestanding BFO by flexoelectric effect is attributed to its tuning on the band diagram across the device, which is illustrated in Fig. 5. The additional built-in electric field generated by the strain gradient couples with the BFO polarization-dependent internal electric field $E_{in}$ to tailor the photovoltaic effect of BFO, which will be discussed in details below.

## Discussion
The fact that bending induced strain gradient changes the photoconductance confirms the contribution of the flexoelectric effect to the photovoltaic effect of freestanding BFO. An additional internal electric field generated by the flexoelectric effect could enhance or weaken the separation of the photoexcited electron–hole pairs. When bending the flexible devices upward as shown in Fig. 5, the thin film undergoes lattice distortions as schematically illustrated in Fig. 1c–d. Since the transferred film is much thinner than the supporter, the in-plane of the film

experiences an increasing tensile strain status along the radius direction, while the out-of-plane of the film is subjected to an increasing compressive strain along the radius direction correspondingly. The in-plane strain in BFO film induced by bending can be roughly calculated using the equation of

$$\varepsilon_{ip} = \frac{\sum_{i=1}^{4} Y_i t_i \left[ \left( \sum_{j=1}^{i} t_j \right) - t_i/2 \right]}{r \sum_{i=1}^{4} Y_i t_i}, \quad (1)$$

$$= \frac{\frac{Y_1 t_1^2}{2} + Y_2 t_2 \left(t_1 + \frac{t_2}{2}\right) + Y_3 t_3 \left(t_1 + t_2 + \frac{t_3}{2}\right) + Y_4 t_4 \left(t_1 + t_2 + t_3 + \frac{t_4}{2}\right)}{r(Y_1 t_1 + Y_2 t_2 + Y_3 t_3 + Y_4 t_4)}, \quad (2)$$

where $t_i$ and $Y_i$ are the thickness and Young's moduli of the $i$-th layer (with sequence of $i$ counted from the top layer), and $r$ is the bending radius of the sample[39]. The thickness of BFO and LSMO film is 100 and 15 nm, respectively; and the thickness of PDMS/PET substrate is ~20 μm. The bending radius used in the bending measurements are from 1.4 to 3.6 mm (Supplementary Fig. 11, Supplementary Table 2). Therefore, by substituting the values of $t_i$, $Y_i$, and $r$, we can estimate the in-plane strain $\varepsilon_{ip}$ in BFO, and the out-of-plane strain $\varepsilon_{op}$ can be obtained by taking account of the Poisson ratio of BFO (−0.3),... The detailed information of how to calculate the strain status is written in Supplementary Table 1 in the Supporting information. The in-plane and out-of-plane strain gradients in BFO thin film can therefore be estimated as $\varepsilon/t_{BFO}$, with the results shown in Fig. 1b. As discussed above, the felxoelectric effect generates an additional built-in electric field that couples with polarization-dependent $E_{in}$ of the flexible device to tune its photovoltaic effect. The flexo-photovoltage ($\Delta V_{oc}$) generated by the flexoelectric effect shown in Fig. 4f can thus be fitted using the following equation:

$$\Delta E_{z,flex} = \lambda \frac{e}{\varepsilon_o a} \frac{d\varepsilon}{dt} \quad (3)$$

where $\lambda$ is a scaling factor close to unity[9], $e$ is the electronic charge, $\varepsilon_o$ is the permittivity of free space, and $a$ is the lattice constant of BFO film[7]. To the best of our knowledge, up to now, there is no report of the precise value of $\lambda$ for BFO. Since $\Delta E_{z,flex}$ can be estimated as $\Delta V_{oc}/t_{BFO}$, the value of $\lambda$ can therefore be back calculated, which is

$$\lambda = \frac{\Delta V_{oc} \varepsilon_o a_{BFO}}{t_{BFO} e SG_{op}}, \quad (4)$$

where $\Delta V_{oc}$ is the flexo-photovoltage shown in Fig. 4f, $t_{BFO}$ is the thickness of BFO film, and $SG_{op}$ is the out-of-plane strain gradient in BFO. By putting the data in Eq. (4), we can obtain the corresponding value of $\lambda$, as shown in Supplementary Table 3. Then, an average value of 0.5 for $\lambda$ is obtained, with the standard deviation of ±0.17. The variation on the value of $\lambda$ for different bending status might result from the experimental error and the soft nature of PDMS that cannot guarantee the precise strain transfer during bending. Using the value of 0.5 for $\lambda$, the flexo-photovoltage generated by the flexoelectric effect can be fitted using the Eq. (3), as shown as in Fig. 4f (green line). The experimental data of $\Delta V_{oc}$ follows the change tread of the fitted data, which justifies our assumption that the photovoltaic effect can be flexoelectrically tuned in the freestanding BFO films, with single domain by simply taking advantage of the mechanical strain.

In this work, the flexoelectric effect along the out-of-plane direction would only be considered, since our devices have the sandwich structure and photovoltaic effect is measured along the out-of-plane direction. The upward bending of the flexible device induces an increasing tensile in-plane strain and an increasing

compressive out-of-plane strain along the radius direction, which can generate an effective downward flexoelectric polarization due to the flexoelectric effect[11]. Consequently, an additional built-in electric field $\Delta E_{z,flexo}$ would couple with the polarization-dependent internal field $E_{in}$ of single-domain BFO to determine its photovoltaic effect, as illustrated in Fig. 5a, b, where the band diagrams of BFO with different polarizations are shown. When BFO polarization points upward, it will be weakened by the downward flexoelectric polarization, and $\Delta E_{z,flexo}$ generated by flexoelectric field will counteract the downward $E_{in}$, resulting in a weaker photovoltaic effect; while when BFO polarization points downward, it will be strengthened by the flexoelectric polarization, and $\Delta E_{z,flexo}$ will reinforce the upward $E_{in}$, leading to steeper band bending of BFO and therefore an enhanced photovoltaic effect by separating more electro–hole pairs. Multistate conductance in freestanding ferroelectric thin film therefore can be realized by mechanical writing, and the reading process is simply done by measuring the photovoltaic response, of which the speed is only limited by the RC-time constant of the circuit. Upward bending was demonstrated only in this work due to the feasibility of our equipment. A flexoelectric polarization pointing upward is anticipated for downward bending that will lead to an opposite change of the photovoltaic response, with the same polarization direction compared with the upward bending. Our results reveal the important role of stain gradient that can be utilized as a new degree of freedom to tune the physical properties of devices. For example, our results indicate that the strain gradient in freestanding ferroelectric thin films can be used to enhance the ferroelectric photovoltaic efficiency. As mentioned in the introduction part, strain gradient has already been proven to be able to mediate the local photoelectric properties of ferroelectric thin films. However, the strain gradient used in their works is either a localized effect or cannot be tuned continuously. From this point of view, our performance metric exceeds the reported approaches, since it is a universal and facile method that can control the strain gradient continuously. In addition to the improvement of ferroelectric photovoltaic outputs, our findings are relevant in strain sensing applications. Our device is able to sense the bending strain, which is more applicable in flexible electronics and gesture recognition. To assess the application metrics of our device as a strain sensor, the Gauge factor (GF) is calculated, which is a characteristic parameter representing the sensitivity of the strain sensors. The GF can be calculated from $\Delta R/(R_0\varepsilon)$, where $\Delta R$ is the resistance change that is equal to $R - R_0$, and $\varepsilon$ is the strain in BFO thin film[40]. In this work, we can use the photoresistance ($V_{oc}/I_{sc}$) to calculate the GF. By calculating the $\Delta R/R_0$ for each bending at different bending cycles, an average value is taken, and GF is finally obtained by $\Delta R/(R_0\varepsilon)$. The calculated GF is listed in Supplementary Table 4. The GF value of our freestanding device is much higher than many of the reported strain sensors[40–43], which indicates its potential applications as a strain sensor with good strain sensitivity. More importantly, our device harvests photon energy and converts it into electrical energy. Therefore, it may be potential for a self-power sensor without external electric power supply.

In summary, continuously strain gradient in a large scale of freestanding BFO thin film can be achieved by simply bending the flexible device. The controllable strain gradient leads to tunable photoconductance in freestanding BFO through the coupling between the additional electric field generated by the flexoelectric effect and the BFO polarization-dependent internal electric field. Consequently, multilevel photovoltage/photocurrent can be mechanically written, while the readout can be optically obtained by measuring the photovoltaic response of the freestanding BFO under light illumination. Moreover, the tuning on the photoconductance by the strain gradient can be repeated well, which

guarantees the functionality of this mechanical tuning method. The demonstrated multiconductance states may find their potential applications in improving the efficiency of photovoltaic devices or as strain sensors. Our findings broaden the horizon of study on the flexoelectric effect in the flexible electronic devices, and might stimulate more research on tuning the physical properties using the strain gradient as the degree of freedom may be stimulated.

## Methods

**Thin film deposition**. SAO sacrificial layer was epitaxially grown on STO single-crystalline substrate by PLD technique at the temperature of 800 °C with the deposition pressure of $10^{-5}$ Torr. Following the growth of SAO, LSMO was deposited at 800 °C with an oxygen partial pressure of 200 mTorr as the bottom electrode, and then BFO was deposited at 700 °C with an oxygen partial pressure of 100 mTorr. The laser energy density used for the film growth was fixed at 1.2 J/cm², and the laser repetition rate for deposition of SAO, LSMO, and BFO thin films was 2, 2, and 5 Hz, respectively.

**Device fabrication**. After the thin film deposition, the sample was immersed into DI water to dissolve the SAO sacrificial layer, and then the residual freestanding LSMO/BFO layer was transferred to Si wafer for the XRD, PFM, and *P–V* loop measurements. To transfer the freestanding oxide heterostructures to flexible substrates, the sample was adhered onto PDMS surface that uses PET as the supporter, and then immersed into DI water. After transferring the freestanding LSMO/BFO heterostructure to supporters, Pt was patterned on top of BFO as the top electrode with the thickness of 10 nm and the size of $50 \times 50$ m² using PLD.

**Materials characterization**. PFM was carried out to measure the polarization of BFO thin film using a commercial atomic force microscope (Asylum Research MFP-3D). Ferroelectric properties and polarization switching were carried out via a commercial ferroelectric tester (Radiant Technologies). Electrical measurements were carried out using a pA meter with direct current voltage source (Hewlett Package 4140B) on a low noise probe station. The light source used in this work was a Halogen lamp of which the energy intensity can be adjusted from 5 to 20 mW/cm².

## Data availability

The data that support the findings of this study are available from the corresponding author upon reasonable request.

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

## Acknowledgements

This work is partially supported by Singapore National Research Foundation (NRF) under CRP Award No. NRF-CRP10-2012-02, the Singapore Ministry of Education

MOE2018-T2-2-043 and MOE AcRF Tier 1- FY2018–P23. L.Y. acknowledges the start up fund from Soochow University, and the support from Priority Academic Program Development (PAPD) of Jiangsu Higher Education Institutions. J.W. acknowledges the start up grant from Southern University of Science and Technology. X.Y. acknowledges the National Natural Science Foundation of China (61674050, 61874158).

## Author contributions

R.G. and J.C. conceived the project. J.C., J.W., and X.Y. supervised the whole project. R.G. fabricated the devices. R.G. and L.Y. carried out the electrical measurements in J.W.'s lab. W. L. and L.Y. participated the data discussion and plotting of the manuscript. A.A. measured the XRD. X.S. carried out the SEM measurements. R.G. and J.C. wrote the manuscript. G.Z., S.C., and L.L. contributed to this project by participating the discussion of the manuscript.

## Competing interests

The authors declare no competing interests.
