## [Peer Review File · Nature Communications]

Reviewers' comments:

Reviewer #1 (Remarks to the Author):

The authors demonstrated continuously tunable photovoltaic responses, via mechanically bending free-standing BiFeO₃ films. They also claimed that such continuous tunability arises from strain gradients and associated flexoelectricity. Flexoelectricity has nowadays attracted much attention, as it occurs in any dielectric and possesses great potential in nanoscale electronic devices. This manuscript suggests a facile method of introducing controllable macro-scale flexoelectric effect, which in turn tunes an important functionality, i.e., photovoltaic response, in oxide thin films. This is interesting and the manuscript is well written, but I still have major concerns as below. These should be addressed before any publication.

(1) Figure 1c,d schematically explains how flexoelectricity contributes to the net internal field, depending on the ferroelectric-polarization (P_{FE}) direction (i.e., upward or downward P_{FE}). However, this explanation is confusing. In terms of "polarization", the upward E_{flexo} corresponds to the upward flexoelectric polarization P_{flexo} . [In fact, the actual physical quantity, generated by a strain gradient, is the flexoelectric polarization P_{flexo} , and the concept of E_{flexo} is just for convenience.] Then, the upward P_{flexo} competes with the downward P_{FE} , not the upward P_{FE} . This is contradictory to what the authors are claiming.

(2) The authors should exclude the effect of strain itself. In the experimental geometry, BiFeO₃ could experience tensile in-plane and compressive out-of-plane strains. In such strained BiFeO₃, the ferroelectric polarization could rotate, resulting in the change of a net out-of-plane polarization. [Note that the polarization in BiFeO₃ (001) films is tilted, consisting of both the out-of-plane and in-plane polarization component.] This might explain the experimental observation, even without consideration of flexoelectricity.

(3) The authors should carefully estimate the strain status in this bending geometry. BiFeO₃ is bent only along a certain in-plane direction, namely, x direction. Then, the strain along the other orthogonal in-plane direction, namely, y direction, would be different from that along x direction. Only the strain along the x direction is experimentally controlled, and the other strains along the y and z directions should be carefully estimated.

(4) It seems that the authors neglected the LSMO and PDMS layers, when they estimate the strain status. How valid is this? In particular, the presence of the intermediate, soft PDMS layer can make it difficult to properly estimate the strains in BiFeO₃ layer.

(5) The internal field could originate from the work-function difference of Pt and LSMO layers, as well as the depolarization field.

Reviewer #2 (Remarks to the Author):

The manuscript of R. Guo et al. presents photoconductance and ferroelectric switching measurements in freestanding thin films of BiFeO₃ while under a bending geometry, which induces a strain gradient. Overall I am rather positive on this type of work – these sorts of flexible films are relatively new, and not much work has been done in this direction which directly utilizes their mechanical flexibility. In my opinion, however, the manuscript in its current form is not at the level of achievement needed for

publication in Nature Communications.

As examples of what would be at the appropriate level, I could imagine 2 straightforward revisions/extensions of this work:

1) From an application point of view, I could not properly envision the possibilities the authors have in mind. The statements such as “Our findings enable ferroelectric thin film based flexible devices to function as multilevel memories and strain sensors” seem vague and do not give the reader a concrete application to consider. For me, the notion of using such macroscopic mechanical bending did not seem practically relevant for memory applications. Perhaps sensors could be relevant. In any case, I would suggest that at least 1 clear application be presented, together with a demonstration of a performance metric with the new structures that exceeds other current approaches. To the extent that the bending response seems to be the focus of the authors, I would then suggest more testing along these lines. For example, Fig. S7 shows extensive fatigue measurements for ferroelectric switching up to 10^5 cycles, but Fig. S8-S10 shows reproducibility only for a few bending cycles (3-10 times). Depending on the application envisioned, I would imagine the many-cycle stability under bending would be quite important to demonstrate, and it is not a priori obvious given the complex LSMO/BFO/Pt structure (I am thinking about dislocations and other mechanical fatigue processes, particularly at the interface with the Pt metal).

2) From a fundamental point of view, I was a bit frustrated by the statement “For perovskite oxide system, λ is known to have the value which is on the order of 10^0 or 10^1 ...” Given that flexoelectricity has already been considered for this material for a number of years, I would imagine there is more literature and measurements on this coefficient. If not however, there is an opportunity here. The authors are ideally poised for a careful measurement of the flexoelectric coupling coefficient given their geometry. Such a measurement, and comparison to the existing literature, would certainly be interesting.

In summary, I feel that the current manuscript presents potentially interesting phenomena, but at a rather shallow level – it seems like a cute demonstration, but it could and should be deeper than that. I have given examples of how this work could be extended more seriously either in terms of applications or fundamental measurements. In my opinion, either is equally fine (both would be outstanding), but such work should be done to raise the impact of these results to the journal standard.

Finally, 3 minor technical points:

1) I don't understand the structure of Fig. 1b – there are 4 points and a continuous line. The text indicates that this is data. Please clarify what is indicated in the figure, and how the strain gradient is independently (?) measured as compared to the radius of curvature.

2) In the section discussing FeRAM and fatigue properties, I think it is relevant to cite related work on using such freestanding films from other groups – I recall experiments using BaTiO₃ from the groups of Alexe and Hwang; there may be others.

3) Author Contributions information. The text here gives very little real information on what most of the authors contributed to the work. To the extent the journal requires this information, this should be properly conveyed.

Reviewer #3 (Remarks to the Author):

The manuscript presents continuously controllable photoconductance of BFO by transferring the free-standing thin films onto soft PDMS. A tunable flexoelectricity degree of freedom is therefore added to the systems, resulting in the continuous tunability of photoconductance and ferroelectric photovoltaic effect. With the flexible capability, the authors are able to create multilevel photoconductance, furthermore, they demonstrate mechanically writing and optically reading methods. In the discussion part, the authors also quantitatively estimate the induced flexoelectricity.

The article is well-written and easy to understand. In my opinion, the presented results are of great importance and of significant novelty, which might meet the general high quality requirement of Nature Communication. However, before I could recommend the acceptance of its publication, there're some issues to be addressed. My comments and concerns are provided as follows:

1. Regarding the growth side, I'm wondering why 4° miscut (001)-STO is chosen for growth. Usually, the increasing vicinity/miscut degrees will affect the surface energy during growth, leading to relatively rough surface. The authors have done a good job in growth side, please elaborate the adoption of the miscut substrate.
2. Did the authors check which type (p-type or n-type) the free standing BFO is? From growth point of view, different growth conditions will lead to different types of BFO films, which could possibly change the results observed by the authors.
3. The authors mentioned the net polarization along out-of-plane is reversed (upward to downward) after free standing process. It seems there's a lack of driving force for the polarization to rotate during the free standing process, please elaborate this observation/result further.
4. P6, line 157, J-V curves under different light intensities were also measured as shown in" Fig. S6a-c", shouldn't it be Fig. S5?
5. P6, line 160, Fig. S6 shall be Fig. S5?
6. P7, line 168, there's no Fig. 3e in the manuscript.
7. P7, line 175, Fig. 3f shall be Fig. 3d?
8. Line 176, "Pt/BFO/LSMO on PDMS retains good photovoltaic property even at 60 °C". One of the major advantages of using oxide-based functional materials is the thermal stability. Could the photovoltaic property measurements go beyond 60 °C? If yes, can the free standing BFO still exhibit excellent multilevel photovoltaic property?
9. The current work focuses on the free standing BFO in upward bending manner, I'm wondering if the observations become different when it comes to "downward" bending condition. The authors don't need to redo all the experiments again, yet some discussion regarding "downward" bending will be helpful.
10. I only see the AFM image of BFO film grown on STO substrate without SAO layer provided in Fig. S4. Though the AFM image of free standing BFO film is shown in Fig. S3, I suggest the authors also include the AFM image of BFO film "with" SAO layer grown on STO substrate in the manuscript. With this the authors can strengthen the non-destructive advantages using water-resolvable SAO sacrificial layers.
11. The authors analyzed the induced flexoelectric field in the discussion section, which I think is of great importance. The interpretation is convincing and supportive by the experimental results. I'm wondering if multilevel conductance behaviors would be significantly affected by the thickness ratio of the BFO film to the PDMS, that is " η " (according to the definition in Line 243). In this manner, there might be more turnabilities while adopting PDMS in different thickness.
12. Some of ferroelectrics are expected to show super elastic behavior in free standing form. Have the authors ever done the experiment with smaller bending radius?
13. From my point of view, P-E data as a functional of bending radius is needed. P-E data while bending might offer strong support for the flexoelectric scenario as discussed by the authors.

Minor:

- i. Line 257, Fig. 4F  4f
- ii. Line 469, (d)  (c)

Reviewer#1

The authors demonstrated continuously tunable photovoltaic responses, via mechanically bending free-standing BiFeO₃ films. They also claimed that such continuous tunability arises from strain gradients and associated flexoelectricity. Flexoelectricity has nowadays attracted much attention, as it occurs in any dielectric and possesses great potential in nanoscale electronic devices. This manuscript suggests a facile method of introducing controllable macro-scale flexoelectric effect, which in turn tunes an important functionality, i.e., photovoltaic response, in oxide thin films. This is interesting and the manuscript is well written, but I still have major concerns as below. These should be addressed before any publication.

Response: We thank very much for the reviewer's positive comments on the novelty and importance of our work. We have carefully addressed the reviewer's concerns as below.

(1) Figure 1c, d schematically explains how flexoelectricity contributes to the net internal field, depending on the ferroelectric-polarization (P_{FE}) direction (i.e., upward or downward P_{FE}). However, this explanation is confusing. In terms of "polarization", the upward E_{flexo} corresponds to the upward flexoelectric polarization P_{flexo} . [In fact, the actual physical quantity, generated by a strain gradient, is the flexoelectric polarization P_{flexo} , and the concept of E_{flexo} is just for convenience.] Then, the upward P_{flexo} competes with the downward P_{FE} , not the upward P_{FE} . This is contradictory to what the authors are claiming.

Response: We thank the reviewer to point out this confusing description in the manuscript. Indeed, the actual physical quantity induced by the strain gradient is the flexoelectric polarization P_{flexo} . However, there is still a lack of consensus on the sign of the flexoelectric coefficients of ferroelectric oxides. Contradictory results have been reported both experimentally and theoretically (*Annu. Rev. Mater. Res.* 43, 387–421 (2013); *Phys. Rev. B* 88, 174107 (2013); *Science* 336, 59 (2012); *Phys. Rev. Lett.* 107, 057602 (2011); *Adv. Mater.* 26(29), 5005 (2014)). Therefore, the direction of P_{flexo} cannot be determined a priori according to the strain gradient direction.

To better clarify this issue, we make an analogy between the defect-dipole polarization and flexoelectric polarization, as shown in **Fig. R1** below. Defect dipoles are prevalent in ferroelectric materials. They are charged defects, which couple to the ferroelectric polarization and result in an imprint field. It is possible to tune the defect-dipole polarization via thermal annealing in pre-poled state (*Adv. Mater.* 26, 5005 (2014)). For example, for as-grown BFO sample on STO which has upward virgin polarization, if we pole it downwards and annealing at high temperature, the defect-dipole polarization P_{defect} will align parallel with the existing

ferroelectric polarization (**Fig. R1a**). The P_{defect} then acts as an effective field that causes the horizontal shift in the P - V loop (**Fig. R1c**). In the meantime, the associated photovoltaic J - V curves (in both P_{up} and P_{down} states) also show corresponding shifts along the voltage axis (**Fig. R1e**). The increase of the J_{sc} and V_{oc} in the P_{down} state can be understood as more effective depolarization field being generated by the P_{defect} .

Turning now to flexoelectric polarization, we found upward bending would lead to similar shift of P - V loop towards positive voltage axis (**Fig. R1d**). Furthermore, the photovoltaic J - V curves exhibit exactly the same trend as the P_{down} annealed sample (**Fig. R1f**). By analogy, we can conclude the upward-bending-induced flexoelectric polarization is also pointing downwards, which helps to enhance the photovoltaic outputs in the P_{down} state (**Fig. R1b**). Our observation is also consistent with other people's report. (*Nat. Nanotechnol.* 10, 972–979 (2015).)

We have modified the manuscript accordingly in the discussion part in page 12 lines 1-12 to make it unambiguous.

Fig. R1. An analogy of P - V loops and J - V curves between defect-dipole polarization and flexoelectric polarization. SG 4 corresponds to the largest strain gradient in this work.

(2) The authors should exclude the effect of strain itself. In the experimental geometry, BiFeO₃ could experience tensile in-plane and compressive out-of-plane strains. In such strained BiFeO₃, the ferroelectric polarization could rotate, resulting in the change of a net out-of-plane polarization. [Note that the polarization in BiFeO₃ (001) films is tilted, consisting of both the out-of-plane and in-plane polarization component.] This might explain the experimental observation, even without consideration of flexoelectricity.

Response: We agree with the reviewer that the polarization of BFO could rotate under the in-plane tensile strain. If this effect dominates, we should observe symmetric changes of the J - V curves in two opposite polarization states, that is, both increase or decrease. However, our experimental results show asymmetric changes of the J - V behaviours. Specifically, the J_{sc} decreased in upward-polarization state, while increased in downward-polarization state when the device was bent. This behaviour cannot be explained by the change of the out-of-plane polarization component, but is in accordance with the flexoelectric effect.

(3) The authors should carefully estimate the strain status in this bending geometry. BiFeO₃ is bent only along a certain in-plane direction, namely, x direction. Then, the strain along the other orthogonal in-plane direction, namely, y direction, would be different from that along x direction. Only the strain along the x direction is experimentally controlled, and the other strains along the y and z directions should be carefully estimated.

Response: We thank the reviewer for pointing out this issue. First, it should be noted that the substrate induced strain was almost fully relieved in the freestanding BFO film, as confirmed by the bulk-like lattice parameter calculated from the XRD result. Now considering a bulk-like rhombohedral structure as shown in **Fig. R2** below, the x, y, z directions, namely, pseudocubic <100> axes, are symmetry-wise equivalent. Consequently, a uniaxial tensile stress along x direction would generate equal strain along y and z directions. The magnitude of the strain can be calculated through the Poisson ratio of the related axes, e.g. $\nu_{zx} = -\frac{\varepsilon_z}{\varepsilon_x} = \nu_{zy} = -\frac{\varepsilon_z}{\varepsilon_y}$. In the case of BFO, ν_{zx} is close to -0.3 according to a previous study (*Phys. Status Solidi A* 214, No.1,1600356 (2017)).

Likewise, we expect similar effect when the strain is applied along y direction from pure symmetry consideration. To verify it, we specially bent a different sample along the y direction with the same bending radius, and we obtained the similar results as bending alone the x direction, as shown in **Fig. R3** below.

Fig. R2. Schematic drawing of the rhombohedral BFO pseudo-cubic cell.

Fig. R3. (a) Change of J - V curves under light illumination with different bending radius along y direction. (b) The corresponding change of V_{oc} and J_{sc} with the in-plane strain gradient. SG1 to SG4 are the increasing 4 strain gradients corresponding to the 4 different bending radii.

(4) It seems that the authors neglected the LSMO and PDMS layers, when they estimate the strain status. How valid is this? In particular, the presence of the intermediate, soft PDMS layer can make it difficult to properly estimate the strains in BiFeO₃ layer.

Response: Thanks for the reviewer's advice. In our previous manuscript, we roughly calculated the strain status in freestanding BFO thin film with referring to Ref. 17 and 20. There are indeed four layers in our devices. Following the reviewer's advice, we recalculate the strain status using a more concrete formula which considers about the whole four layers. The details about how we calculate the strain status are written below:

Since there are total 4 layers, the most complete expression is:

$$\varepsilon_{ip} = \frac{\sum_{i=1}^4 Y_i t_i \left[\left(\sum_{j=1}^i t_j \right) - t_i / 2 \right]}{R \sum_{i=1}^4 Y_i t_i}$$

(1)

$$= \frac{\frac{Y_1 t_1^2}{2} + Y_2 t_2 \left(t_1 + \frac{t_2}{2} \right) + Y_3 t_3 \left(t_1 + t_2 + \frac{t_3}{2} \right) + Y_4 t_4 \left(t_1 + t_2 + t_3 + \frac{t_4}{2} \right)}{R(Y_1 t_1 + Y_2 t_2 + Y_3 t_3 + Y_4 t_4)}$$

(2)

where t_i and Y_i are the thicknesses and Young's Moduli of the i -th layer (with sequence of i counted from the top layer), while R is the radius of bending curvature.

[Ref: *Adv. Energy Mater* 7, 1700535 (2017)]

Note that for two layers (including substrate), this expression restores back into the form of

$$\varepsilon_{ip} = \frac{t_f + t_s}{2R} \left[\frac{1 + 2\eta + \chi\eta^2}{(1 + \eta)(1 + \chi\eta)} \right]$$

(3)

(where $\eta = \frac{t_f}{t_s}$ and $\chi = \frac{Y_f}{Y_s}$), which was used in our former manuscript.

[Ref: *Sci. Adv.* 3, e1700121 (2017); *Adv. Mater.* 29, 1702411 (2017).]

The following parameters (from top to bottom) are used for calculation:

1st layer is BFO, $t_1 = t_{\text{BFO}} = 100$ nm, $Y_1 = Y_{\text{BFO}} = 144$ GPa

[Ref: *Appl. Phys. Lett.* 102, 182905 (2013)]

2nd layer is LSMO, $t_2 = t_{\text{LSMO}} = 15$ nm, $Y_2 = Y_{\text{LSMO}} = 500$ GPa

[Ref: *Ultrasonics* 44, e1223, (2006)]

3rd layer is PDMS, $t_3 = t_{\text{PDMS}} \sim 5$ μm , $Y_3 = Y_{\text{PDMS}} = 3$ MPa

[Ref: *J. Appl. Polym. Sci.* 131, 41050 (2014)]

4th layer is PET, $t_4 = t_{\text{PET}} \sim 15$ μm , $Y_4 = Y_{\text{PET}} = 3.5\text{-}11$ Gpa (A medium value of 7.5 Gpa was used here.)

<https://www.makeitfrom.com/material-properties/Polyethylene-Terephthalate-PET-PETE>

After substituting all the values into the formula, the in-plane strain in BFO $\varepsilon_{ip} = 1.057 \times 10^{-5} / R$ can be obtained, which is slightly larger compared with the data calculated previously ($\varepsilon_{ip} = 0.922 \times 10^{-5} / R$). We therefore use equation (1) in the new manuscript. By putting the value of R , in-plane strain gradient can be obtained, and the out-of-plane strain gradient can be obtained by taking account of the Poisson ratio of BFO (-0.3), $\nu = -\frac{\varepsilon_{op}}{\varepsilon_{ip}}$. The Poisson ratio for BFO film is -

0.3. The detailed information on how to calculate the strain status are written in the Supporting information **Table S1** in page 39 and the main text in page 10, respectively.

Table S1. The calculated strain gradient and the fitted flexo-voltage (ΔV_{oc}).

Bending radius (m)	In-plane strain gradient (m^{-1})	Out-of-plane strain gradient (m^{-1})	Fitted $\Delta E_{z,flex}$ (V/m)	Fitted $\Delta V_{oc} = \Delta E_{z,flex} * t_{BFO}$ (V)
0.0036	29360.83057	-8808.24917	-202013.267	0.02021
0.0028	37749.63931	-11324.89179	-259731.344	0.02598
0.0019	55631.0474	-16689.31422	-382761.98	0.03828
0.0014	75499.27862	-22649.78359	-538900.12	0.05389

At last, we agree that soft PDMS may cause the calculation of strain status not precise, we therefore emphasize that the strain status was roughly estimated in the manuscript. This might be one of the reasons that causes the deviation between the estimated and experimentally measured flexo-photovoltage.

(5) The internal field could originate from the work-function difference of Pt and LSMO layers, as well as the depolarization field.

Response: Yes, we agree with the reviewer. The polarization dependent photovoltaic effect indicates it is mainly resulted from the depolarization field. However, the asymmetric V_{oc} and I_{sc} under positive and negative polarization directions is likely due to the different work functions of the top and bottom electrodes, which also generates an internal field that does not depend on the polarization direction. We have added this explanation in page 6 line 15-17 highlighted with red.

Reviewer#2

The manuscript of R. Guo et al. presents photoconductance and ferroelectric switching measurements in freestanding thin films of BiFeO₃ while under a bending geometry, which induces a strain gradient. Overall I am rather positive on this type of work – these sorts of flexible films are relatively new, and not much work has been done in this direction which directly utilizes their mechanical flexibility. In my opinion, however, the manuscript in its current form is not at the level of achievement needed for publication in Nature Communications.

Response: Thanks very much for the reviewer’s positive comments on the novelty of our work. Following the reviewer’s constructive advice, we have done more experiments and made a major revision to address the reviewer’s concerns.

As examples of what would be at the appropriate level, I could imagine 2 straightforward revisions/extensions of this work:

(1) From an application point of view, I could not properly envision the possibilities the authors have in mind. The statements such as “Our findings enable ferroelectric thin film based flexible devices to function as multilevel memories and strain sensors” seem vague and do not give the reader a concrete application to consider. For me, the notion of using such macroscopic mechanical bending did not seem practically relevant for memory applications. Perhaps sensors could be relevant. In any case, I would suggest that at least 1 clear application be presented, together with a demonstration of a performance metric with the new structures that exceeds other current approaches.

Response: We agree that the macroscopic mechanical bending did not seem practically relevant for memory application. Therefore, we have deleted the application as multilevel memory in manuscript to make it more rigorous. Based on our results, we think the strain gradient in freestanding ferroelectric thin films can be used to enhance the ferroelectric photovoltaic efficiency. As mentioned in the introduction part, strain gradient has already been proven to be able to mediate the local photoelectric properties of ferroelectric thin films. However, the strain gradient used in their works is obtained by controlling the film growth or applying an external force via a SPM tip, which is either a localized effect or cannot be tuned continuously. From this point of view, our performance metric with the new structures exceeds the reported approaches, since it is a universal and facile method which can control the strain gradient continuously.

In addition to the improvement of ferroelectric photovoltaic outputs, our findings are relevant in strain sensing applications, as suggested by the reviewer. Our device is able to sense the bending strain, which is more applicable in flexible electronics and gesture recognition. To assess the application metrics of our device as a strain sensor, we calculate the Gauge factor, which is a characteristic parameter representing the sensitivity of the strain sensors. The gauge factor (GF) can be calculated from $\Delta R/(R_0 \epsilon)$, where ΔR is the resistance change which is equal to $R - R_0$, and ϵ is the strain in BFO thin film.[*Adv. Mater. Technol.* 4, 1900578 (2019)] In this work, we use the photo-resistance (V_{oc}/I_{sc}) to calculate the GF. By calculating the $\Delta R/R_0$ for each bending at different bending cycles, an average value is taken, and GF is finally obtained by $\Delta R/(R_0 \epsilon)$. The calculated GF is listed in the **Table S4** below. The GF value of our freestanding device is much higher than many of the reported strain sensors,[*Adv. Mater. Technol.* 4, 1900578 (2019); *Nano*

Let. 10, 490 (2010); *ACS Nano* 8, 5154 (2014); *Adv. Mater.* 26, 2022 (2014)] which indicates its potential applications as a strain sensor with very good strain sensitivity. More importantly, our device harvests photon energy and converts it into electrical energy. Therefore, it may be potential for a self-power sensor without external electric power supply.

We have modified the discussion part accordingly in the main text which includes the potential applications of our flexible devices in page 12 lines 19-25 and page 13 lines 1-16, and added **Table S4** in the supporting information in page 40-41.

Table S4. Calculated gauge factor (GF) based on the change of the photo-resistance due to bending. ϵ_{ip} is the strain in freestanding BFO film corresponding to four different bending radii in the study.

ϵ_{ip}	0.29%	0.37%	0.55%	0.75%
$\Delta R/R_0=(R-R_0)/R_0$	0.105	0.138	0.215	0.29
$GF=\Delta R/R_0\epsilon$	36.2	37.3	39.1	38.6

To the extent that the bending response seems to be the focus of the authors, I would then suggest more testing along these lines. For example, Fig. S7 shows extensive fatigue measurements for ferroelectric switching up to 10^5 cycles, but Fig. S8-S10 shows reproducibility only for a few bending cycles (3-10 times). Depending on the application envisioned, I would imagine the many-cycle stability under bending would be quite important to demonstrate, and it is not a priori obvious given the complex LSMO/BFO/Pt structure (I am thinking about dislocations and other mechanical fatigue processes, particularly at the interface with the Pt metal).

Response: Yes, the bending response is the focus of this work. Following the reviewer’s advice, we have done more bending testing cycles with the results shown in **Fig. 4e** and **Fig. S10**. After bending the flexible device to different cycles, the response of photovoltaic property to different bending radius was measured to test the reproducibility of the flexible device. Besides, *J-V* switching curves were also measured after bending the device for different cycles, as shown in **Fig. S9**. These results confirms the good reproducibility and stability of our freestanding device. Furthermore, we have also done the fatigue and retention measurements for the device in the bending status, as shown in **Fig. S7c-d**, which indicates that the freestanding devices can sustain up to 100k cycles for both flat and bending status. The added experimental data are put in **Fig. 4e-f**, **Fig. S7c-d**, **Fig. S9**, and **Fig. S10**, respectively.

Fig. 4. Bending-tunable photovoltaic properties of freestanding Pt/BFO/LSMO. (e) The change of the V_{oc} and J_{sc} with the function of the introduced strain gradient (SG) after different bending cycles. The polarization of BFO is downward. (f) The experimental data of ΔV_{oc} (flexo-photovoltage) with both upward and downward polarization directions, and the fitted ΔV_{oc} as the function of the strain gradient by using the value of 0.5 for scaling factor λ . SG1 to SG4 corresponds to the increasing strain gradients of four different bending radii in Fig. 1b.

Fig. S9. Photovoltaic and J - V switching properties of freestanding Pt/BFO/LSMO after multiple bending. (a) J - V curves under light illumination (with light intensity of $20 \text{ mW}/\text{cm}^2$) and in dark for both polarizations of the flexible Pt/BFO/LSMO device after bending 300 times. (b) J - V switching loops in dark of the flexible Pt/BFO/LSMO device after bending 300 times.

Fig. S10. Reproducibility of the tunable photovoltaic properties of freestanding Pt/BFO/LSMO. (a-g) J - V curves under light illumination (with light intensity of 20 mW/cm^2) after 2nd, 3rd, 5th, 10th, 50th, 100th, 300th bending, respectively. After the bending, the response of the photovoltaic property to different bending radii was tested. J - V curve within a very small voltage range is measured to read the change of V_{oc} and J_{sc} in response to different bending radii. (h) The corresponding change of the V_{oc} and J_{sc} with the function of the introduced in-plane strain gradient (SG) after different bending cycles. The polarization of BFO is upward. SG1 to SG 4 corresponds to the four different radii of the bending in Fig. 1b.

Fig. S7. Fatigue and retention results. (a) and (b) Fatigue and retention properties of the freestanding device in flat status for both polarizations. (c) Fatigue and (d) retention performances of the freestanding device in bending status for both polarizations. For the fatigue measurements, the J - V curves were measured after switching the polarization using $\pm 3V$, with the pulse width of 1ms. The freestanding BFO devices show good retention properties. Besides, the freestanding devices can sustain 100k cycles for both flat and bending status.

(2) From a fundamental point of view, I was a bit frustrated by the statement “For perovskite oxide system, lambda is known to have the value which is on the order of 10^0 or 10^1” Given that flexoelectricity has already been considered for this material for a number of years, I would imagine there is more literature and measurements on this coefficient. If not however, there is an opportunity here. The authors are ideally poised for a careful measurement of the flexoelectric coupling coefficient given their geometry. Such a measurement, and comparison to the existing literature, would certainly be interesting.

Response: Thanks for the reviewer’s constructive suggestion. λ is a scaling factor close to unity (Ref. 9 in manuscript; *Appl. Phys. Lett.* 81, 3440 (2002)). However, up to now, to our best of knowledge, the precise value of λ for BFO thin film is still not known yet. In our former manuscript, we used the value of 1 for λ by referring to Prof. T. W. Noh’s group work (Ref. 7 in manuscript). In their work, they pointed it out that λ for BFO is not known, and they used the

value of 1 in their work. Likewise, in our former manuscript, by assuming $\lambda=1$, we calculated the flexoelectric field, which agrees with the shifts in photovoltaic open-circuit voltage (V_{oc}).

We agree that an experimental data for the value of λ is important, and λ can be experimentally calculated through the equation of

$$E_{z,\text{flex}} = \lambda \frac{e}{\epsilon_0 a} \frac{d\epsilon}{dt}, \quad (\text{Ref. 7 in manuscript}) \quad (1)$$

where e is the electronic charge, ϵ_0 is the permittivity of free space and a is the lattice constant of BFO film. Since the flexoelectric field $E_{z,\text{flex}}$ can be estimated as $\Delta V_{oc}/t_{\text{BFO}}$, we can therefore back calculate the value of λ , which is

$$\lambda = \frac{\Delta V_{oc} \epsilon_0 a_{\text{BFO}}}{t_{\text{BFO}} e SG_{op}}, \quad (2)$$

where ΔV_{oc} is the flexo-voltage shown in **Fig. 4f**, t_{BFO} is the thickness of BFO film, SG_{op} is the out-of-plane strain gradient in BFO. By substituting the data in equation (2), we can obtain the corresponding value of λ , as shown in **Table S3**. Then, an average value of 0.5 for λ is obtained, with the standard deviation of ± 0.17 . The calculated λ from our experimental results is close to unity 1. The variation on the value of λ for different bending status might result from the experimental error and the soft nature of PDMS which cannot guarantee the precise strain transfer during bending.

Table S3. The calculated value of λ corresponding to each ΔV_{oc} in Fig. 4f.

ΔV_{oc}	0.012	0.018	0.032	0.053	0.0141	0.0344	0.056	0.07
λ	0.29701	0.34651	0.41801	0.51104	0.34899	0.66222	0.73153	0.67377

To clearly demonstrate the point, we have written down the detailed information of how we calculate λ and fitted our experimental data in the main text in page11 lines 8-21 as marked in red, and added **Table S3** in the supporting information in page 40.

In summary, I feel that the current manuscript presents potentially interesting phenomena, but at a rather shallow level – it seems like a cute demonstration, but it could and should be deeper than that. I have given examples of how this work could be extended more seriously either in terms of applications or fundamental measurements. In my opinion, either is equally fine (both would be

outstanding), but such work should be done to raise the impact of these results to the journal standard.

Response: Thanks very much for the reviewer’s recognition of the importance of our work and the constructive suggestions on how to improve the impact of our work. Following the reviewer’s advice, we have extended the content of our manuscript both in terms of application of the devices and the fundamental measurement of λ . After the major revision, we believe our new manuscript is at the level of achievement needed for publication in Nature Communications.

Finally, 3 minor technical points:

1) I don’t understand the structure of Fig. 1b – there are 4 points and a continuous line. The text indicates that this is data. Please clarify what is indicated in the figure, and how the strain gradient is independently (?) measured as compared to the radius of curvature.

Response: We are sorry for the confusing description. The solid line (in-plane strain gradient) in the previous manuscript is the calculated strain gradient as a function of curvature radius according the equation (1) in the main text. In the revised manuscript, out-of-plane strain gradient is also added in **Fig. 1b** which is calculated according to $\nu = -\frac{\epsilon_{op}}{\epsilon_{ip}}$, where Poisson ratio ν of BFO is taken by -0.3. The 4 solid points on the line are the 4 bending curvatures conducted in our experiments. In this work, we only change the bending radius and all strain gradient are calculated from the bending radius.

Fig. 1 (b). The solid lines are the calculated strain gradient as a function of the curvature radius. The two groups of 4 solid points are the in-plane and our-of-plane strain gradients corresponding to the different bending levels of the freestanding devices conducted in this work.

2) In the section discussing FeRAM and fatigue properties, I think it is relevant to cite related

work on using such freestanding films from other groups – I recall experiments using BaTiO₃ from the groups of Alexe and Hwang; there may be others.

Response: Thanks very much for the reviewer's suggestion. We are sorry that we forgot to cite the relative two papers from the groups of Prof. Alexe and Hwang. We have already cited them (Ref. 33 and 34) in page 4 lines 2-3 as marked in red color.

3) Author Contributions information. The text here gives very little real information on what most of the authors contributed to the work. To the extent the journal requires this information, this should be properly conveyed.

Response: Thanks for the reviewer's reminder. We have given more detailed information on the contribution part as marked in red.

Reviewer #3

The manuscript presents continuously controllable photoconductance of BFO by transferring the free-standing thin films onto soft PDMS. A tunable flexoelectricity degree of freedom is therefore added to the systems, resulting in the continuous tunability of photoconductance and ferroelectric photovoltaic effect. With the flexible capability, the authors are able to create multilevel photoconductance, furthermore, they demonstrate mechanically writing and optically reading methods. In the discussion part, the authors also quantitatively estimate the induced flexoelectricity.

The article is well-written and easy to understand. In my opinion, the presented results are of great importance and of significant novelty, which might meet the general high quality requirement of Nature Communication. However, before I could recommend the acceptance of its publication, there're some issues to be addressed. My comments and concerns are provided as follows:

Response: We thank very much for the reviewer's highly recognition of the importance and writing of our work, and the corrections on the minor mistakes. Following the reviewer's advice, we have addressed the raised issues to improve the quality of our manuscript.

1. Regarding the growth side, I'm wondering why 4° miscut (001)-STO is chosen for growth. Usually, the increasing vicinity/miscut degrees will affect the surface energy during growth, leading to relatively rough surface. The authors have done a good job in growth side, please elaborate the adoption of the miscut substrate.

Response: We chose 4° miscut (001)-STO to grow single-domain BFO, in order to eliminate complications from multiple domain variants. As shown in **Fig. R1**, the miscut edge can lift the degeneracy of the multiple domains in BFO, resulting in preferable growth of single domain variant (*Adv. Mater.* 21, 1-7 (2009); *Appl. Phys. Lett.* 99, 052903 (2011)). It is true that the increasing vicinity/miscut degrees can lead to relatively rough surface compared with BFO thin film grown on flat STO, however, it will not affect our experiments since the switchable polarization dependent photovoltaic effect is the key issue in this work. Following reviewer's advice, we have elaborated the reason of choosing miscut STO substrate in page 4, lines 6-8.

Fig. R1. (a-c) Schematic pictures showing the preferable growth of BFO domain variants on different vicinal (miscut) substrates. (d) Relevant lateral and vertical strains imposed by the vicinal substrates.

2. Did the authors check which type (p-type or n-type) the free standing BFO is? From growth point of view, different growth conditions will lead to different types of BFO films, which could possibly change the results observed by the authors.

Response: We agree with the review that different growth conditions might lead to different types of BFO films. For example, oxygen vacancies in BFO might lead to n-type behavior (*Adv. Mater.* 23, 1277 (2011)), while Bi-deficiency might lead to p-type (*Adv. Func. Mater.* 22(5), 1040 (2012)). In this work, we did not check which type the free-standing BFO is, since our main point is to demonstrate the photovoltaic property of the free-standing BFO. Besides, the leakage current of our device at small voltage range is around the magnitude of 10^{-11} A, indicating very low carrier concentration of our BFO film.

3. The authors mentioned the net polarization along out-of-plane is reversed (upward to downward) after free standing process. It seems there's a lack of driving force for the

polarization to rotate during the free standing process, please elaborate this observation/result further.

Response: There is some misunderstanding here. The free-standing process does not change the polarization direction of BFO. BFO grown on SAO layer has a downward out-of-plane polarization, which is opposite to BFO grown on STO without SAO layer (upward out-of-plane polarization). We assumed this is because the grown SAO layer leads to a different termination of the following LSMO layer, which therefore induces different polarization directions of the subsequent ferroelectric BFO layer (Ref.35 and 36 in manuscript). After transferring, free-standing BFO thin film has the same downward out-of-plane polarization as the as-grown BFO/LSMO/SAO/STO film. In order to make the point clearer, we have added the PFM images of the as grown BFO/LSMO/SAO/STO film, as shown in **Fig. S4** in page 32.

Fig. S4. PFM results of as-grown BFO/LSMO/SAO/STO sample. (a) Topography of as-grown BFO film. (b) Out-of-plane PFM image with the virgin state and the state after switching using a bias of -5 and +5 V, respectively. (c) The corresponding in-plane PFM image. The PFM results reveal that the as-grown BFO film has a virgin downward single domain structure. The freestanding BFO has the same polarization direction as the as-grown one with SAO layer.

4. P6, line 157, J-V curves under different light intensities were also measured as shown in” Fig. S6a-c”, shouldn’t it be Fig. S5?

Response: Thanks for the reviewer’s correction. We are sorry for the mistyping and have corrected it already.

5. P6, line 160, Fig. S6 shall be Fig. S5?

Response: Thanks for the reviewer’s correction. We are sorry for the mistyping and have corrected it already.

6. P7, line 168, there’s no Fig. 3e in the manuscript.

Response: Thanks for the reviewer's correction. It should be Fig. 3c instead. We are sorry for our careless mistake.

7. P7, line 175, Fig. 3f shall be Fig. 3d?

Response: Yes, it should be Fig. 3d. Sorry for the mistake. We have already corrected it.

8. Line 176, "Pt/BFO/LSMO on PDMS retains good photovoltaic property even at 60 °C". One of the major advantages of using oxide-based functional materials is the thermal stability. Could the photovoltaic property measurements go beyond 60 °C? If yes, can the free standing BFO still exhibit excellent multilevel photovoltaic property?

Response: *J-V* curves at different temperatures were measured to test the basic thermal stability of our flexible devices. We found that Pt/BFO/LSMO on PDMS still retains good photovoltaic property at 60 °C. However, as we continue to increase the temperature, we found that the PDMS starts to harden and deform. The hardening problem of PDMS becomes very serious and almost lost the flexibility at 80 °C, so we only measure the photovoltaic property up to 60 °C to prove it has thermal stability. It has been reported that the ferroelectricity of free-standing ferroelectric thin films on mica can be maintained at the temperature up to 175 °C (Ref.17 in manuscript). We therefore believe that with flexible substrates which can hold higher temperature, the thermal stability measurements can go beyond 60 °C.

9. The current work focuses on the free standing BFO in upward bending manner, I'm wondering if the observations become different when it comes to "downward" bending condition. The authors don't need to redo all the experiments again, yet some discussion regarding "downward" bending will be helpful.

Response: Thanks for the reviewer's understanding. Upward bending was demonstrated only due to the feasibility of our equipment. Following the reviewer's advice, we have added the discussion of "An effective flexoelectric polarization pointing up is anticipated for downward bending which will lead to an opposite change of the photovoltaic response with the same polarization direction compared with the upward bending.", as marked in red in page 12, lines 15-19.

10. I only see the AFM image of BFO film grown on STO substrate without SAO layer provided in Fig. S4. Though the AFM image of free standing BFO film is shown in Fig. S3, I suggest the authors also include the AFM image of BFO film "with" SAO layer grown on STO substrate in the manuscript. With this the authors can strengthen the non-destructive advantages using water-resolvable SAO sacrificial layers.

Response: Following the reviewer’s advice, we have added the AFM and PFM images of BFO film with SAO layer grown on STO, as shown in **Fig. S4** in page 32.

11. The authors analyzed the induced flexoelectric field in the discussion section, which I think is of great importance. The interpretation is convincing and supportive by the experimental results. I’m wondering if multilevel conductance behaviors would be significantly affected by the thickness ratio of the BFO film to the PDMS, that is “ η ” (according to the definition in Line 243). In this manner, there might be more tunabilities while adopting PDMS in different thickness.

Response: Thanks very much for the reviewer’s positive comments. Based on the formula which was used to calculate the strain, we agree with the reviewer that there might be more tunabilities while adopting PDMS in different thickness. To emphasize this point, we have added one more sentence in page 39 by saying “Since BFO layer is much thinner than the substrate, choosing a thinner flexible substrate can increase the minimum bending radius and therefore more tunabilities can be realized.”

12. Some of ferroelectrics are expected to show super elastic behavior in free standing form. Have the authors ever done the experiment with smaller bending radius?

Response: Yes, very recently, ferroelectric BaTiO₃ thin films are reported to show fantastic super elastic behavior in free standing form (*Science*, 366, 475 (2019)). In our study, we have also found the rolling of free standing BFO, like the image in the right side in Fig. 2 (b). However, in our work, we did not do the experiment with smaller bending radius, since we had already observed the multiple photoconductance via four different bending levels, which has already proven our point. We think the reviewer gives us very good suggestion on our future work, we can study other application of free-standing BFO based on the recently reported super elastic BaTiO₃. We have added “super-elastic ferroelectric single-crystal membrane have been demonstrated very recently” in page 4 line 1-2, and cited this paper as **Ref. 32** in our manuscript.

13. From my point of view, P-E data as a functional of bending radius is needed. P-E data while bending might offer strong support for the flexoelectric scenario as discussed by the authors.

Response: Following the reviewer’s advice, we have measured the *P-V* loop with the function of bending radius, as shown in **Fig. 4b**. *P-V* loops show a horizontal right-shift with the increase of the strain gradient, which indicates an effective built-in electric field generated by flexoelectric effect. We have added the content in page 8 line 21-24.

Fig. 4b. P - V loops of the flexible device as a function of the bending radius. SG 1 to SG 4 corresponds to the four strain gradients caused by four levels of bending.

Minor:

i. Line 257, Fig. 4F  4f

Response: Thanks for the correction. We have corrected it.

ii. Line 469, (d)  (c)

Response: Thanks for the correction. We have corrected it.

REVIEWERS' COMMENTS:

Reviewer #1 (Remarks to the Author):

I read through the authors' response to all comments carefully, and I am satisfied with their response as well as their revision. So, I recommend the publication of this manuscript.

Reviewer #2 (Remarks to the Author):

Firstly, I apologize to the authors and the editor for the delayed review - this certainly does not reflect my appraisal of the manuscript.

I have carefully reviewed the response and revisions. Overall I feel the authors seriously considered and tried to address all of the comments from all reviewers, which were quite extensive. I also feel the manuscript is stronger with the revisions, and will be an important paper in the field.

As such, I fully recommend acceptance, with no need to further review.

Let me just mention 2 points which the authors may optionally consider addressing if they easily can, or so choose:

1) I think there are some signs of 'fatigue' in the now extended bending tests. This is not a criticism for this paper, which is an early exploration of this interesting direction. Even if speculative, it might be worthwhile if the authors suggest what might be the origin - mechanical fatigue (plastic deformation), or ferroelectric domain rotation, etc. - simple to guide the reader for what aspects might be the best to focus on for further development towards applications.

2) The work is presented assuming that bending and illumination do not effect the ferroelectric domain configuration, which is assumed to be a single macroscopic domain. To the extent the authors feel that this is conclusive, it might be worth mentioning and substantiating.

Reviewer #3 (Remarks to the Author):

In this revision, the authors have addressed all my questions and concerns properly. Now I can recommend its publication in Nature Communications.

Reviewer #1 (Remarks to the Author):

I read through the authors' response to all comments carefully, and I am satisfied with their response as well as their revision. So, I recommend the publication of this manuscript.

Reply: Thanks very much for the reviewer.

Reviewer #2 (Remarks to the Author):

Firstly, I apologize to the authors and the editor for the delayed review - this certainly does not reflect my appraisal of the manuscript.

I have carefully reviewed the response and revisions. Overall I feel the authors seriously considered and tried to address all of the comments from all reviewers, which were quite extensive. I also feel the manuscript is stronger with the revisions, and will be an important paper in the field.

As such, I fully recommend acceptance, with no need to further review.

Reply: Thanks very much for the reviewer.

Let me just mention 2 points which the authors may optionally consider addressing if they easily can, or so choose:

1) I think there are some signs of 'fatigue' in the now extended bending tests. This is not a criticism for this paper, which is an early exploration of this interesting direction. Even if speculative, it might be worthwhile if the authors suggest what might be the origin - mechanical fatigue (plastic deformation), or ferroelectric domain rotation, etc. - simple to guide the reader for what aspects might be the best to focus on for further development towards applications.

Reply: Thanks for the reviewer's suggestion. We have added one sentence in the manuscript in page 8 line 12-13 which suggests the possible reasons for the fatigue.

2) The work is presented assuming that bending and illumination do not effect the ferroelectric domain configuration, which is assumed to be a single macroscopic

domain. To the extent the authors feel that this is conclusive, it might be worth mentioning and substantiating.

Reply: Yes, its single domain. We used PUND to verify there is no partial or complete switching of the polarization upon bending or illumination. To make it clearer, we emphasized single domain in the manuscript as marked in red.

Reviewer #3 (Remarks to the Author):

In this revision, the authors have addressed all my questions and concerns properly. Now I can recommend its publication in Nature Communications.

Reply: Thanks very much for the reviewer.